# Non-Optic Glioma-like Lesions in Adult Neurofibromatosis Type 1 Patients

**DOI:** 10.3390/diagnostics15010067

**Published:** 2024-12-30

**Authors:** Walter Taal, Bart Zick, Bart J. Emmer, Martin J. van den Bent

**Affiliations:** 1Department of Neuro-Oncology/Neurology, Erasmus MC Cancer Institute, 3015 GD Rotterdam, The Netherlands; 2Department of Radiology, Erasmus MC, Dr. Molewaterplein 40, 3015 GD Rotterdam, The Netherlands

**Keywords:** neurofibromatosis type 1, NF1, glioma, adults, cohort study

## Abstract

**Background/Objectives:** Physicians face clinical dilemmas in the diagnosis of non-optic intraparenchymal lesions on MRI brain scans of patients with neurofibromatosis type 1. As the incidence and evolution of these lesions into adulthood remain unclear, we conducted a retrospective study on this topic. **Methods:** All adult neurofibromatosis type 1 patients who had at least one MRI brain scan in our center were selected for this study. Brain lesions with contrast enhancement after gadolinium administration and/or mass effect were named “glioma-like lesions”. **Results:** In our cohort of 396 adult neurofibromatosis type 1 patients, 182 had at least one MRI scan of the brain. A total of 48 glioma-like lesions were found in 38/182 patients. The majority of glioma-like lesions remained stable, decreased in size or even disappeared during a median follow-up time of 8.5 years. Twelve glioma-like lesions in 11/182 patients were resected or biopsied, and histology showed gliomas of astrocytic origin (WHO grade 1–4). **Conclusions:** It was concluded from these data that asymptomatic glioma-like lesions on MRI brain scans in neurofibromatosis type 1 patients, either with contrast enhancement and/or mass effect, had an indolent nature. Mildly symptomatic or asymptomatic patients can therefore be followed without invasive diagnostic and therapeutic procedures.

## 1. Introduction

Neurofibromatosis type 1 (NF1) is one of the most common neurocutaneous disorders with an incidence of approximately 1 in 2000–3000 individuals [1]. NF1 patients carry a highly increased risk of developing low-grade gliomas, with pilocytic astrocytoma (WHO grade 1) being the most common [2,3,4]. High-grade gliomas are also more frequently found in NF1, especially in adults [5]. Up to 20% of patients with NF1 have an optic pathway glioma (OPG), which is typically a pilocytic astrocytoma [6,7,8]. OPGs rarely develop or require medical intervention after the age of seven [8,9,10]. Pilocytic astrocytomas have also been described in other cerebral regions in NF1 patients [10,11]. NF1-related low-grade gliomas have a relatively favorable prognosis compared to that of non-NF1-associated low-grade gliomas [4,12,13]. Because of this favorable prognosis, it has been postulated that some space-occupying intraparenchymal lesions are in fact so-called focal areas of signal intensity (FASIs), also known as unidentified bright objects (UBOs), especially as spontaneous regression or even remission has been observed [14].

FASIs are hyperintensities on T2-weighted MRI brain images and are mostly observed in the basal ganglia, cerebellum, thalamus, brainstem and subcortical white matter. As per their definition, FASIs are not space-occupying and do not show contrast enhancement on MRI scans. FASIs are pathognomonic in NF1 and usually regress with age [15,16,17,18]. The differentiation between low-grade gliomas and FASIs on MRI scans is sometimes difficult and the clinical management of T2-weighted hyperintensities in NF1 patients may be challenging. Moreover, gadolinium-enhancing lesions which spontaneously regress or even vanish have been described in NF1 patients, adding even more confusion to this subject [19].

Since these intraparenchymal lesions may present diagnostic challenges and may lead to unnecessary invasive diagnostic and therapeutic approaches, a better understanding of their clinical behavior is necessary. We therefore retrospectively studied the morphology and evolution of non-optic intraparenchymal lesions into adulthood in NF1 patients in order to better understand the nature of these lesions and to develop a rational approach towards them.

## 2. Methods

**Patients**. In this retrospective longitudinal study, we analyzed all adult patients (≥18 years of age) with NF1, known at the ENCORE-NF1 clinic at Erasmus Medical Centre in Rotterdam, the Netherlands. All patients who underwent at least one available MRI brain scan at an adult age (≥18 years of age) between 2003 and 2017 were selected for this study. In these patients, all MRIs were studied, including those that were made before these patients had reached the age of 18 years old. The present study was conducted according to local and national regulations and was approved by the Institutional Review Board of the Erasmus Medical Center. We collected general information, such as age, gender, NF1 criteria and medical history.

For inclusion, MRI scans needed to contain at least the following sequences: an axial T1-weighted scan with and without gadolinium enhancement and an axial T2-weighted scan. Gadolinium administration was not mandatory if there were no lesions detected on MRI. Each MRI brain study was analyzed for its indication and presence of additional sequences. Additional sequences such as fluid-attenuated inversion recovery (FLAIR), diffusion-weighted imaging (DWI) with an apparent diffusion coefficient (ADC) and perfusion imaging were not required, but, if available, they were also analyzed in order to expand or support the (differential) diagnosis. MRI scans were analyzed by two investigators (B.Z. and B.E., a board-certified neuroradiologist).

Each MRI scan was analyzed for the presence of intracranial abnormalities and general findings. Optic pathway lesions were not taken into account. Non-optic intraparenchymal lesions were divided into the following:I.FASIs: all T2-weighted/FLAIR lesions without contrast enhancement and no mass effect; these were scored as being present or not.II.“Glioma like lesions” (GLLs): Lesions exhibiting clear mass effect and/or contrast enhancement and/or being in any MRI brain scan during the follow-up. These lesions were considered as possibly having a neoplastic origin and were described more precisely, with respect to intensity on T1-weighted and T2-weighted sequences; location; aspect (cystic, necrotic, presence of blood); and morphology (circumscribed, diffuse, homo- or heterogeneous enhancement) both before and after contrast administration, atrophy and gray or white matter involvement. Furthermore, these lesions were measured in two directions on the axial T2-weighted sequence (maximal diameter with perpendicular diameter in mm, exclusive of any cystic, necrotic or bloody components).

All GLLs were measured to visualize relevant changes during the follow-up if available. We considered an increase in size significant if the increase was at least 25%, based on its two-dimensional measurement on axial T2-weighted images. A decrease in size was considered significant if the decrease was at least 25%. Complete remission was defined if the lesion disappeared completely from both T1- and T2-weighted MRI images during follow-up.

Descriptive statistics were obtained for various characteristics of the subjects and the frequencies of MRI brain findings. For database collection and analysis, Microsoft Excel 2013 (version number 15.0) and Microsoft Access 2013 (version number 15.0) and SPSS version 23.0.1 were used.

## 3. Results

All 396 adult NF1 patients visiting the ENCORE-NF1 Center in Rotterdam at least once were reviewed. One hundred and eighty-two NF1 patients fulfilled the criteria and had at least one MRI of the brain after the age of 18 years old. In 127/182 (69%) patients, the first MRI was made under the age of 18 years. In total, 4 of the 182 patients did not have a T1-weighted sequence with gadolinium enhancement. However, these patients did not have intraparenchymal findings on the other MRI sequences for which contrast administration was necessary to differentiate between the three categories of lesions in this study. From these 182 NF1 patients, a total of 546 MRIs of the brain were available for review.

Table 1 shows the characteristics of the 182 NF1 patients with at least one MRI brain scan at adult age. Males and females were equally present, and most patients had prominent phenotypic NF1 characteristics such as (sub)cutaneous neurofibromas, CAL or freckling. Headache and fatigue were the most frequently spontaneously reported symptoms. Headache was mostly classified as tension-type or medication-induced/dependent due to analgesics. The most common neurological deficit associated with an intracranial lesion was a motor and/or sensory deficit. In 15/182 (8%) NF1 patients, a malignancy was found, of which the malignant peripheral nerve sheath tumor was the most common (6%, Table 1).

In 56/182 patients, the indication for a brain MRI scan included neurological signs or symptoms (e.g., motor loss, pathological reflexes, ataxia or bulbar signs) or seizure(s). In 126/182 patients, MRI scans were made because of routine screening purposes or minor complaints without alarming signs (such as headache and concentration deficits).

Non-optic intraparenchymal lesions, including FASIs and GLLs, were found in 123/182 (68%) patients during follow-up (see Table 2). In total, 12 out of the 182 patients had multiple sclerosis-like lesions such as multiple lesions adjacent to the ventricles, including ovoid lesions perpendicular to the ventricles and involvement of the corpus callosum or juxtacortical lesions. Two of these patients had clinically established multiple sclerosis (see Table 1).

In 38/182 patients, a total of 48 GLLs were found (see Table 2). Twenty-nine GLLs showed contrast enhancement at least on one MRI scan and nineteen GLLs showed mass effect without contrast enhancement. More than half of the GLLs were located supratentorial, mostly in the corpus callosum and frontal lobe. The infratentorial GLLs were mainly located in the pons and brachium pontis (see Table 3).

The median follow-up duration in patients with a GLL was 9.7 years [range 0–20.8]. The product of the largest diameter and the perpendicular diameter of all GLLs in square cm was plotted against the patient’s age (Figure 1). Follow-up scans were available in 36/38 patients with 46 GLLs: 6/46 GLLs increased in size, 24/46 GLLs remained stable, 12/46 GLLs decreased in size (Figure 2) and 4/46 GLLs disappeared completely (including 2/46 GLLs with contrast enhancement). In total, 7 of the 29 enhancing GLLs showed spontaneous disappearance of contrast enhancement, including 2/29 GLLs that disappeared completely. The contrast-enhancing part of all other enhancing GLLs remained stable or decreased in size (Figure 3).

Twelve GLLs in 11/182 (6%) patients were resected or biopsied (Table 4). Three GLLs were resected because of radiological progression only (patient 1–3). All other lesions were discovered due to neurological signs and/or symptoms (patient 4–11). Histology showed gliomas of astrocytic origin (WHO grade I–IV) but was inconclusive in three patients (patient 4–6). Four patients (including the three patients with radiological progression only) did not receive postoperative treatment and remained clinically and radiologically stable during follow-up (patient 1–3 and 5; range of follow-up: 67–298 months). Seven patients (with eight gliomas) received postoperative radiotherapy, which in two cases was combined with temozolomide chemotherapy (patient 4 and 10).

## 4. Discussion

Our study found a very high incidence of non-optic intracranial lesions (including glioma-like lesions and FASIs) in 123/182 (68%) NF1 patients. Even more strikingly, GLLs with alarming features, such as mass effect and/or contrast enhancement, were found in 38/182 (21%) NF1 patients and, notably, many of these GLLs remained stable or decreased in size (36/46 GLLs; 78%) or even disappeared (4/46 GLLs; 9%) and, regularly, enhancement was temporary or disappeared with time (7/29 enhancing GLLs; 24%). None of these cases were clinically symptomatic. Almost half of all 396 adult NF1 patients (182/396; 46%) visiting the ENCORE-NF1 clinic received an MRI scan of the brain, indicating a low threshold for scanning, often because of the follow-up of lesions discovered in their youth. Most MRI scans in NF1 patients (126/182; 69%) were made because screening purposes, follow-up or minor complaints such as headaches, and only a third of the scans (56/182; 31%) were made because of focal neurological deficits with suspicion of an intracranial cause.

The retrospective design of this study has its limitations with respect to the MRI findings. Firstly, no strict follow-up MRI protocol was used and the total number of MRI scans and their frequency were variable between but also in individual patients. Consequently, changes in the reported lesions may have been missed in some cases, and the incidence of asymptomatic increasing and of stable, decreasing or disappearing lesions as reported in this study could be an underestimation of the true incidence. Secondly, in 214/396 (54%) adult NF1 patients that visited our outpatient clinic, MRI scans of the brain were not performed at adult age. As adult NF1 patients without any MRI scan did not have any clinical reason for an MRI scan (i.e., MRI findings in childhood or neurologic deficits), it is likely that the incidence of MRI findings in this study is an overestimation when projected on the total cohort of 396 adult NF1 patients. Still, even if all these 214/396 patients had normal MRI findings, the total number of adult NF1 patients with abnormal MRI scans would be remarkable, with 123/396 (31%) patients with intracranial lesions, including 38/396 (9.6%) with GLLs.

No clear definition of FASIs exists in the literature [20]. In accordance with previous studies, we defined FASIs as lesions which never exhibited any mass effect or contrast enhancement during follow-up [21]. In 98/182 patients with MRI scans, FASIs were identified. Ten patients (two patients with established multiple sclerosis) had small FASIs showing similarities with multiple sclerosis lesions. Interestingly, a relation between multiple sclerosis and NF1 has been suggested in the literature [22,23,24,25]. The prevalence of 0.5% (2 out of 396 NF1 patients) in our total cohort of NF1 patients is higher than one would expect in the Netherlands (prevalence of multiple sclerosis in the Netherlands is 0.1%), although coincidence could play a role with these small numbers.

The benign natural clinical course of FASIs has been studied in NF1 children [18,26,27]. One on these studies showed a decrease in the prevalence of FASIs with increased age in NF1 children: about 80% of the NF1 children at age 2–4 had FASIs but only about 20% at age 16–18 [18]. Only one study on the histology of FASIs in children has been performed in which myelinopathy was regarded as the main explanation for this phenomena [28]. In FASI vacuolar changes, spongiform myelopathy, atypical glial infiltrates, perivascular gliosis and microcalcifications were seen histologically. The high signal intensity on T2-weighted MRI images seemed to be caused by the high-water content vacuoles. It has been postulated that through this way it is also possible for FASIs to disappear with time on MRI [28,29,30].

One or more asymptomatic GLLs were found in 38/182 (21%) patients during follow-up. This finding seems to be relatively high in comparison with other studies on NF1 children and adults. These studies also show that the NF1 population has an increased risk of developing gliomas, with a rate of (radiologically suspected) gliomas varying between 1.43% and 4.78% [2,5,29,31,32,33,34,35,36]. A study by Sellmer et al. on non-optic gliomas in NF1 children and adults reported non-optic gliomas in 24/562 (4.3%) patients, of which nine were histologically proven [37]. A partial explanation for our high rate of GLLs compared to that of this study could be the unselected patient population of Sellmer et al., but as stated above even if all our 214/396 patients without an MRI brain scan had normal MRI findings, this would still have led to 38/396 (9.6%) patients with GLLs in our cohort. Furthermore, Sellmer et al. used different selection criteria for their non-optic glioma. Our criteria for a GLL were mass effect and/or enhancement as these are alarming signs for the clinician. Sellmer et al. also included the location, size and evolution over time of the lesion, although they did not further specify these criteria. Notably, in this respect, no regressing “gliomas” were found in the study of Sellmer et al. If we took out all regressing, stable or temporarily enhancing lesions in our study, this would have led to a much lower percentage of patients with GLLs, but this is why we speak of GLLs (glioma-like lesions), contrasting the term “non-optic glioma” used in the study of Sellmer et al. [37].

This raises the question of if all GLLs in our study are in fact gliomas. The differential diagnosis for these GLLs, besides atypical FASIs, are demyelinating diseases (e.g., multiple sclerosis), vascular lesions (ischemia or small bleedings), epileptic-related lesions, infection, inflammation or metastasis. However, evidence for a non-malignant origin of the enhancing GLLs was lacking in our cohort as the clinical and radiological evolution was not compatible with these diagnoses. The observation of prolonged clinically asymptomatic lesions and even the decrease in size or complete disappearance of lesions also pleads against multiple sclerosis, infection, stroke, high-grade gliomas or metastases. The radiological findings and indolent clinical behavior of the GLLs best fit the diagnosis of an atypical FASI or low-grade glioma.

The histology of all resected GLLs showed a glioma of astrocytic lineage: 4/12 were pilocytic astrocytomas (WHO grade 1), 1/12 was a pilomyxoid astrocytoma (WHO grade 3), 1/12 was an anaplastic astrocytoma (WHO grade 3), 3/12 were glioblastomas (WHO grade 4) and 3/12 were astrocytomas with an inconclusive WHO grade. This is more or less in accordance with the literature. In a series of 100 gliomas in NF1 patients of all ages, histology was reported in 32 patients: twenty-three pilocytic astrocytomas, five glioblastomas, three gangliogliomas and one pilomyxoid astrocytoma (formerly graded as WHO grade II) [38]. Histology in the series of Sellmer et al. showed that 8/9 were pilocytic astrocytomas and 1/9 was a dysembrioplastic neuroepithelial tumor [37]. In the series of Gutmann et al., histology showed that 3/15 were pilocytic astrocytomas, 6/15 were WHO grade II astrocytomas, 4/15 were WHO grade III astrocytomas and 2/15 WHO were grade IV astrocytomas [5].

The recent ERN GENTURIS guideline states that screening for gliomas with MRI should be considered at the age of transition from childhood to adulthood for all patients with NF1 and incidentally detected GLLs should be followed up with imaging [39]. Furthermore, this guideline suggests not to treat asymptomatic or mildly symptomatic GLLs [38,40]. Our results support these recommendations.

The likelihood of diagnosing a diffuse glioma (WHO grade 2–4) diminishes if the lesion does not increase over time. Moreover, previous research showed that GLLs rarely increase significantly beyond 5 years of follow-up [37,38]. Consequently, it appears justified to follow patients with a confirmed GLL with MRI for at least 5 years (Figure 4).

## Figures and Tables

**Figure 1 diagnostics-15-00067-f001:**
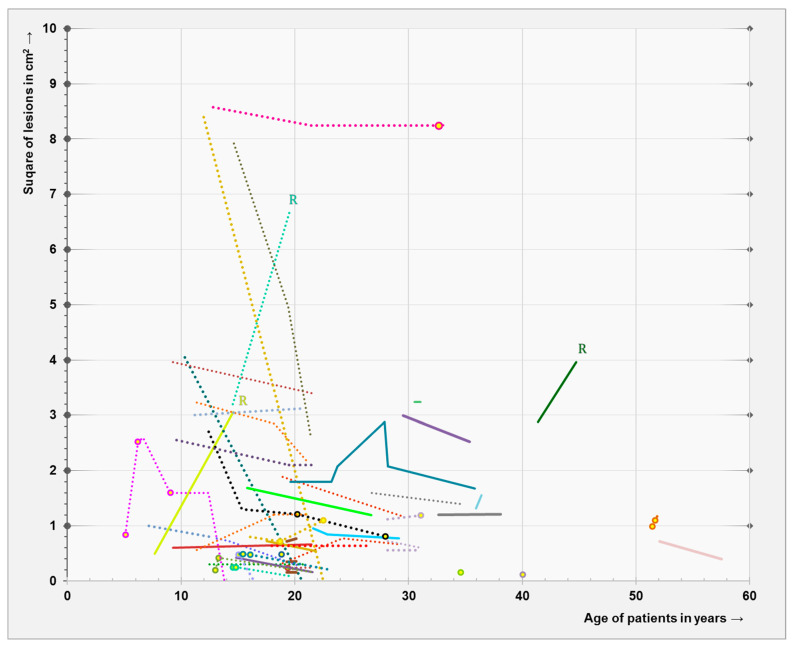
The evolution of asymptomatic glioma-like lesions in NF1 patients. A total of 48 asymptomatic glioma-like lesions, defined as MRI lesions exhibiting clear mass effect and/or contrast enhancement, were found in 38 out of 182 adult NF1 patients, with at least 1 MRI scan of the brain after the age of 18 years old. The size of every lesion is measured on axial T2-weighted MRI brain images, and the product of the largest diameter and the perpendicular diameter is plotted against the patient’s age. Lines with the same color represent lesions within the same patient. Solid lines present enhancing lesions. Dotted lines represent non-enhancing lesions, but the open balls in these dotted lines represent temporary contrast enhancement of the same lesion. The 3 R’s mean that the lesions have been resected because of fast radiological progression (see Table 4; patient 1–3).

**Figure 2 diagnostics-15-00067-f002:**
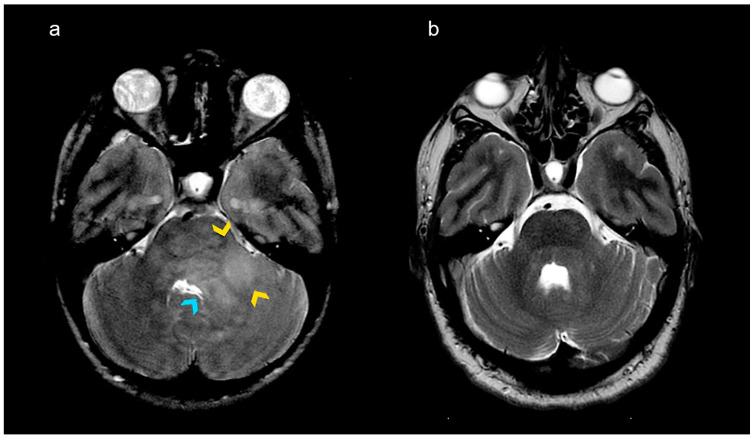
**A non-enhancing space-occupying glioma-like lesion (GLL).** (**a**) A GLL in the left cerebellar peduncle in an 8-year-old neurofibromatosis type 1 patient. The MRI was performed because of a mild ataxia of the left extremities. The MRI shows a space-occupying lesion with high signal on the T2-weighted images (the yellow arrowheads point to the diffuse hyperintense lesion, while the blue arrowhead points to compressions of the 4th ventricle from the left side). The lesion in the cerebellar peduncle was regarded as a low-grade glioma. The lesion was followed up frequently and no treatment was given. The patient’s symptoms improved within years, as the lesion and its space-occupying appearance were also disappearing. (**b**) MRI of the same patient at the age of 22 years old shows a significant decrease in the T2 lesion and its mass effect.

**Figure 3 diagnostics-15-00067-f003:**
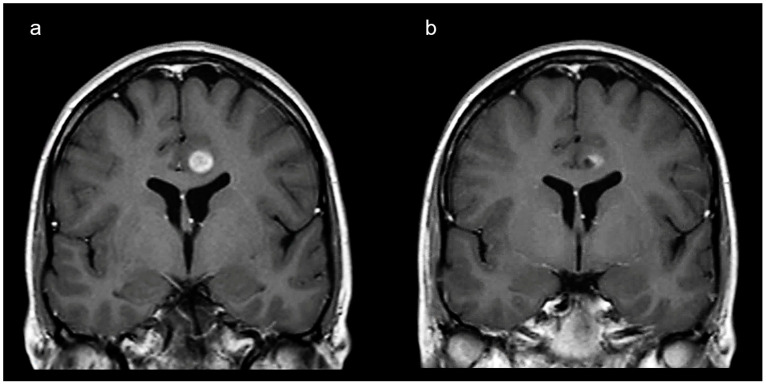
An enhancing glioma-like lesion. (**a**) The brain MRI shows a lesion in in the left frontal cerebral lobe of a 26-year-old neurofibromatosis type 1 patient. The MRI scan was performed because of screening purposes and the patient did not have neurological signs or symptoms. The lesion shows some mass effect and clear contrast enhancement. This lesion was characterized as a low-grade glioma (most probably a pilocytic astrocytoma). (**b**) The brain MRI at the age of 30 years shows a reduction in size of the enhancing part of the lesion, without any intervention.

**Figure 4 diagnostics-15-00067-f004:**
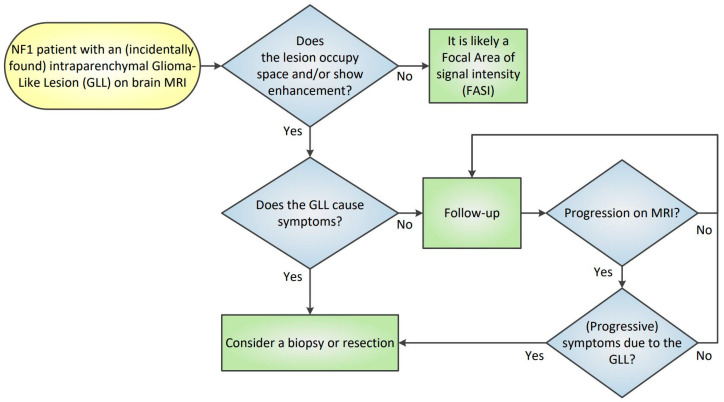
**Diagnostic flowchart for glioma-like lesions.** Focal areas of signal intensity (FASIs) appear hyperintense on T2-weighted MRI and do not exhibit space-occupying characteristics or enhancement after intravenous contrast administration. FASIs are typically asymptomatic and tend to decrease in number and size with advancing age. Consequently, there is no requirement for repeat brain MRI in cases of FASIs. Conversely, gliomas always present as space-occupying lesions. Post-contrast enhancement on brain MRI scans can suggest either a pilocytic astrocytoma (WHO grade 1) or a high-grade glioma (WHO grade 3 or 4). It is recommended that mildly symptomatic or asymptomatic glioma-like lesions (GLLs), with mass effect and/or contrast enhancement, undergo MRI follow-up for a minimum duration of five years. Patients exhibiting symptoms of a GLL should be evaluated in a multidisciplinary meeting to determine the need for biopsy or resection.

**Table 1 diagnostics-15-00067-t001:** Characteristics of all neurofibromatosis type 1 patients with at least one MRI scan of the brain at adult age.

Patient Characteristics	
Median age at time of last MRI scan	31 [range 18–68] years
	Number of patients (%)
Number of patients	182
Male	87 (48%)
Female	95 (52%)
Neurofibroma(s)	
Cutaneous	152 (84%)
Subcutaneous	28 (15%)
Superficial plexiform	49 (27%)
Deep plexiform	56 (31%)
No	20 (11%)
Unknown	4 (2%)
Café-au-lait spots	
≥6	135 (74%)
<6	30 (16%)
No	7 (4%)
Unknown	10 (5%)
Freckling	
Yes	127 (70%)
No	24 (13%)
Unknown	31 (17%)
Most frequent spontaneously reported symptoms	
Headache	82 (45%)
Fatigue	47 (26%)
Blurred vision	23 (13%)
Attention/concentration problems	15 (8%)
Most frequent neurological deficits associated with an intracranial cause	
Any	60 (33%)
Motor and/or sensory deficit(s)	38 (21%)
Decreased visual acuity	25 (14%)
Seizure(s)/epilepsy	24 (13%)
Binocular diplopia	7 (4%)
Ataxia	7 (4%)
Malignancies	15 (8%)
Malignant peripheral nerve sheath tumor	6 (3%)
with intracranial leptomeningeal metastases	1 (0.5%) ^a^
Gastrointestinal stromal tumor	3 (2%)
Breast cancer	2 (1%)
Ovarian cancer	1 (0.5%)
Non-small-cell lung cancer	1 (0.5%)
with intracerebral metastases	1 (0.5%) ^a^
Pheochromocytoma	4 (2%)
Non-malignancies	
Multiple sclerosis	2 (1%)
Histologically proven WHO grade 1 meningioma	3 (2%)

^a^ These intraparenchymal lesions were not taken into account as glioma-like lesions (GLLs).

**Table 2 diagnostics-15-00067-t002:** Intraparenchymal findings in neurofibromatosis type 1 patients with at least one MRI brain scan at adult age.

	Number of Patients (%)	Number of Lesions
Patients with at least one MRI brain scan at adult age	182	
Total number of MRI scans	546	
Patients with		
Any intraparenchymal lesion	123 (68%)	
Focal areas of signal intensity (FASIs) *	98 (54%)	
Multiple sclerosis-like lesions	12 (7%)	
Glioma-like lesions (GLLs) **	38 (21%)	48
GLLs with enhancement	26 (14%)	29
GLLs with only	15 (8%)	19

* Focal areas of signal intensity (FASIs) were defined as T2-weighted hyperintense intraparenchymal lesions without mass effect and/or contrast enhancement (during follow-up). ** Glioma-like lesions (GLLs) were defined as intraparenchymal lesions that showed contrast enhancement and/or mass effect on at least one MRI scan during follow-up.

**Table 3 diagnostics-15-00067-t003:** Location of glioma-like lesions (GLLs) on the MRI brain scan of neurofibromatosis type 1 patients.

Location	All GLLs	*Enhancing GLLs*	*Non-Enhancing GLLs*
Supra- and infratentorial	48	29	19
Supratentorial	28	19	9
Frontal lobe	7	5	2
Parietal lobe	4	4	
Temporal lobe	1	1	
Occipital lobe	2	1	1
Basal ganglia (putamen)	1	1	
Corpus callosum	10	6	4
Paraventricular	3	1	2
Infratentorial	20	10	10
Mesencephalon	3	3	
Pons/brachium pontis	14	5	9
Fourth ventricle	2	2	
Cerebellum	1		1

Abbreviations: GLL: glioma-like lesion.

**Table 4 diagnostics-15-00067-t004:** Clinical features of NF1 patients with histologically proven intraparenchymal lesions.

Patient	M/F	Age (years)	Reason for Intervention	Histology	Aspect on MRI Scan	Location	Treatment	Follow-Up	Outcome
1	M	15	Radiological progression	Pilocytic astrocytoma	Enhancing lesion of 17 × 19 mm, without cysts or mass effect	Left cerebellum	Complete resection	98 months	No recurrence
2	M	19	Radiological progression	Pilocytic astrocytoma	Non-enhancing T2 lesion (23 × 29 mm) with mass effect	Left frontal lobe	Partial resection	58 months	No recurrence
3	M	41	Radiological progression	Pilocytic astrocytoma	Partially enhancing lesion (31 × 23 mm), with cysts	Fourth ventricle	Partial resection	67 months	Stable disease
4	M	28	Ataxia, motor deficit	High-grade astrocytoma (inconclusive) MGMT+	Lesion with high T2 signal (99 × 33 mm), with multifocal enhancing lesions with central necrosis	Cervical spine, medulla oblongata and pons	Partial resection, radiotherapy and temozolomide chemotherapy	11 months	Stable disease
5	F	9	Nystagmus	Probable pilocytic astrocytoma (inconclusive)	Non-enhancing T2 lesion (46 × 49 mm) with mass effect.	Right cerebellum and brachium pontis	Biopsy	289 months	Stable disease
6	M	57	Balance problems	Astrocytic glioma (at least WHO grade II)	Enhancing and partially cystic lesion (52 × 27 mm) without necrosis, hydrocephalus	Cerebellar vermis	Partial resection and radiotherapy	9 months	Deceased
7	M	50	Headache, left-sided hemiparesis and frontal syndrome	Pilomyxoid astrocytoma (WHO grade III)	Enhancing lesion (43 × 43 mm) with central necrosis, hydrocephalus	Right thalamus	Partial resection, radiotherapy, partial re-resection	14 months	Deceased
8	M	27	Progressive headache	Anaplastic astrocytoma (WHO grade III)	Homogenous enhancing lesion (38 × 28 mm), without necrosis, hydrocephalus	Right thalamus	Biopsy, radiotherapy	53 months	Deceased
9	F	30	Progressive headache, left hemiparesis	Glioblastoma (WHO grade IV)	Enhancing lesion (32 × 43 mm) with central necrosis	Right frontal lobe	Complete resection, radiotherapy, re-resection	31 months	Deceased
10	M	47	Balance problems, hemianopia	Glioblastoma (WHO grade IV)	Enhancing lesion with necrosis	Left parieto-occipital lobe	Partial resection, chemo-irradiation with temozolomide	2 months	Stable
11	F	16	Hemiataxia	3 × pilocytic astrocytoma (WHO grade I)	First scans are missing	Left cerebellum	Partial resection, re-resection, radiotherapy and re-re-resection Biopsy, no therapeutic options	120 months	Stable
		39	Amnesic syndrome, progressive headache	Glioblastoma (WHO grade IV)	Enhancing lesion (12 × 18 mm), with central necrosis	Right hippocampal region		4 months	Deceased

Abbreviations: M: male; F: female; MGMT+: hypermethylation of the O6-methylguanine-DNA-methyltransferase (MGMT) gene.

## Data Availability

The data presented in this study are available in anonymized form on request from the corresponding author. The data are not publicly available due to protect study participant privacy, because of the fact that the data is not anonymized.

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
