# Peer review of "Non-Optic Glioma-like Lesions in Adult Neurofibromatosis Type 1 Patients"

_diagnostics, 2024, doi:10.3390/diagnostics15010067_

Round 1
Reviewer 1 Report
Comments and Suggestions for Authors
Taal et al. report on non-optic glioma-like lesions in adult NF1 patients. The authors had a median follow up time of 8.5 years. Majority of the lesions remained stable or shrunk in size or disappeared. This is a well written and concise articles.
These glioma like lesions are major challenge in managing NF1 patients. The data from this study shows that majority of these lesions that are asymptotic or mildly asymptomatic can be monitored with serial MRI without need for biopsy or any intervention.
Comments
1) Figure 1: Find that figure 1 doesn't clearly depict the change in size of the lesions over time. Recommend a SPIDER plot that shows % change from baseline on the y axis and time interval on the x axis. This may make it easier to interpret the figure.
2) Can the authors include a clinical decision algorithm that they would suggest for management of the lesions. A figure depicting the decision tree may be helpful.
Author Response
Comment 1: Figure 1: Find that figure 1 doesn't clearly depict the change in size of the lesions over time. Recommend a SPIDER plot that shows % change from baseline on the y axis and time interval on the x axis. This may make it easier to interpret the figure.
Response 1: We acknowledge the reviewer's concern that Figure 1 may be somewhat challenging to interpret in terms of the changes in lesion size over time. However, a spider plot would not convey the size of the lesions (with some of the largest lesions demonstrating the most significant volume reduction) or the patients' ages. Therefore, we prefer to retain the current figure.
Comment 2: Can the authors include a clinical decision algorithm that they would suggest for management of the lesions. A figure depicting the decision tree may be helpful.
Response 2: We concur with the reviewer and have created a decision flow-chart (figure 4+ legend), which we hope will be included in the manuscript.
Reviewer 2 Report
Comments and Suggestions for Authors
The authors are to be commended as their work addresses a highly interesting phenomenon, namely the evolution of T2 lesions in the brain with or without contrast enhancement into adulthood in NF-1 patients
The observation that such lesions can remain stable or even disappear over time, raises a lot of unanswered questions.
The most important of these is, why a true glioma, an invasive tumor of the brain, should not grow over time. The paper cannot provide an answer here.
Almost all of the biopsy-proven gliomas (6% GLL were biopsied) in this cohort were subsequently treated (10 of 11), and follow-up period in some is relatively short (< 14 months) in 5 of 11 patients. Five patients (mostly higher grade gliomas) have died.
As a conclusion, GLL as well as FASI in NF1 patients could be considered as non-malignant and transient lesions of the brain (with as yet undefined aetiology), whereas true (malignant) gliomas need to be detected by special MRI methods or biopsies and still require follow-up or treatment.
In my opinion, there is no evidence of spontaneous and permanent disappearance of a confirmed invasive glioma in this cohort.
I agree with the authors that most “GLL and FASI” in NF1 patients are benign and may disappear over time
Author Response
Comment 1: In my opinion, there is no evidence of spontaneous and permanent disappearance of a confirmed invasive glioma in this cohort.
Response 1: We appreciate the reviewer's constructive feedback and concur with their comments. As a result, we have added the following section at the end of the discussion: “The likelihood of diagnosing a diffuse glioma (WHO grade 2-4) diminishes if the lesion does not increase over time. Moreover, previous research showed that GGL rarely increase significantly beyond 5 years of follow-up. Consequently, it appears justified to follow patients with a confirmed GGL with MRI for at least 5 years.”
Furthermore we have included a flowchart (figure 4).